# Development and Preliminary Application of a Droplet Digital PCR Assay for Quantifying the Oncolytic Herpes Simplex Virus Type 1 in the Clinical-Grade Production

**DOI:** 10.3390/v15010178

**Published:** 2023-01-07

**Authors:** Miaomiao Guo, Li Deng, Hongyang Liang, Yuyao Du, Wenrui Gao, Na Tian, Ying Bi, Jinghua Li, Tiancong Ma, Yuntao Zhang, Hui Wang

**Affiliations:** 1Beijing Institute of Biological Products Company Limited, 100176 Beijing, China; 2China National Biotec Group Company Limited, 100024 Beijing, China

**Keywords:** droplet digital PCR, quantification, oncolytic virus (OV), oncolytic herpes simplex virus (oHSV), virus production

## Abstract

Oncolytic herpes simplex virus (oHSV) is a type of virus that selectively targets and kills cancer cells, leaving normal cells unharmed. Accurate viral titer is of great importance for the production and application of oHSV products. Droplet digital PCR (ddPCR) is known for having good reproducibility, not requiring a standard curve, not being affected by inhibitors, and being precise even in the detection of low copies. In the present study, we developed a droplet digital PCR assay for the quantification of HSV-1 and applied it in the oHSV production. The established ddPCR showed good specificity, linearity, a low limit of quantification, great reproducibility, and accuracy. The quantification result was well-associated with that of plaque assay and CCID_50_. Amplification of the purified virus without DNA extraction by ddPCR presented similar results to that from the extracted DNA, confirming the good resistance against PCR inhibitors. With the ddPCR, viral titer could be monitored in real time during the production of oHSV; the optimal harvest time was determined for the best virus yield in each batch. The ddPCR can be used as a useful tool for the quantification of oHSV and greatly facilitate the manufacturing process of oHSV products.

## 1. Introduction

Herpes simplex virus types 1 and 2 (HSV-1, -2) are ubiquitous and important human pathogens. According to the WHO report [1], an estimated 3.7 billion people under the age of 50 (67%) have an HSV-1 infection globally, and 491 million people aged 15~49 (13%) worldwide have an HSV-2 infection. Nevertheless, in the last three decades, a vast amount of interest has been placed on the development of HSV as an oncolytic virus (OV) to treat cancer. In fact, HSV-1 has been one of the best and most widely used OVs around the world, which is the viral backbone for the only FDA-approved OV product, Imlygic, developed by Amgen [2].

For oncolytic virus development and production, the accurate quantification of the virus is extremely critical because the quality of OV products is vitally associated with the titer, and they usually exert the therapeutic effects in a dose-dependent manner in both pre-clinical and clinical use [3]. Although polymerase chain reaction (PCR) is recognized as the standard method for the sensitive and specific diagnosis of HSV infections in clinical infection, it is not good for the quantitative analysis [4]. Traditional plaque assay is recognized as the golden standard method for measuring viral titers, including for HSV [5]. However, there are significant limitations of this method, such as (1) long hands-on time during the experiment and the waiting time for the results (usually 3~7 days), (2) cumbersome procedures with a low throughput when titrating multiple clones, and (3) the inherent subjectivity and possible errors during the plaque counting [6]. 50% cell culture infectious dose (CCID_50_) is also widely used for virus titration [7].

Quantitative PCR (qPCR) is well-established for measuring viral nucleic acid by providing absolute quantification of a genomic target when used with an appropriate standard of known concentration. It has been broadly used for virus detection and the quantification of the viral genomic copy number for numerous viruses, due to its rapid detection, high sensitivity, high throughput, and cost-saving by the reduction of labor and materials [8]. A few different real-time PCR assays for the detection of HSV have been developed [9,10,11]. However, their drawbacks should also be considered. A calibration curve is needed for the precise quantification of viral nucleic acid by qPCR. In addition, qPCR is susceptible to inhibitors, which leads to variable qualitative and quantitative results when detecting different types of samples, such as human fluids, swab samples, tissues, cell culture supernatants, and so forth [12]. In this regard, it is of great significance to explore better methods for HSV detection and quantification.

Droplet digital PCR (ddPCR) is an emerging quantitative method (known as the third generation of qPCR) that allows absolute quantification of the target nucleic acids by fractionating the sample into multiple parallel PCR reactions [13,14]. Every droplet in the ddPCR is independent. The ddPCR is nanoPCR, and the detection is from one nanoliter reverse micelle (water in oil) with a fluorescent probe. The target nucleic acids contain zero, one, or more copies in each droplet [15]. This process does not use the calibration curves [16]. The absolute number of target nucleic acids in the original sample can be calculated using Poisson statistics from the ratio of positive to total partitions [17,18,19]. Compared to the qPCR, ddPCR has a few advantages. Of the advantages, the most attractive one is that ddPCR detects multiple targets with high accuracy and specificity without the need to generate standard curves. The specificity of conventional qPCR can be affected by sample impurities, due to the nonspecific binding of dyes, while ddPCR avoids the problem by separating the reactions into many droplets [20,21]. In addition, ddPCR shows an outstanding reproducibility, as it is much less affected by inhibitors than qPCR. It is also highly reliable and can yield precise results, even in the detection of low copies. Moreover, qPCR requires standards for quantification, which makes the results easily affected by noise between sample runs and standard runs [22]. DdPCR has been proved to be a sensitive, repeatable, and rapid method used in research [23] and field studies [24]. As a result, ddPCR has been substantially applied into the quantification of the gene copy number [25], clinical diagnoses of various microorganisms’ infections [26,27], industrial production [28], and a variety of research as well [29,30]. DdPCR is a more mature dPCR technology that has been widely used in medical investigations and clinical applications [31,32].

oHSV has great potential in the treatment of cancer as an immunotherapeutic drug. One big challenge for oHSV development is the large-scale production of high-titer virus. In the present study, we established a ddPCR assay for the detection and quantification of HSV-1 and applied it into the virus titration for different processing stages during oHSV production, which demonstrated viral titer results comparable with the results of traditional plaque assay and CCID_50_. The specificity, lower limit of quantification, linearity, and reproducibility were well-verified as well. The established ddPCR assay substantially facilitated the rapid virus titration, as well as the whole oHSV production process.

## 2. Materials and Methods

### 2.1. Virus and Cells

The wild-type herpes simplex virus type 1 (wt HSV-1), isolated from a clinical swab sample, and the oncolytic HSV (oHSV) were obtained from Dr. Hongkai Zhang at the State Key Laboratory of Medicinal Chemical Biology at Nankai University. The HSV-2 genome was kindly provided by the Wuhan Institute of Virology, Chinese Academy of Sciences. The varicella zoster virus genome was extracted from the virus stock in our company. Virus-producing Vero cells, obtained from WHO, were cultured in DMEM (Gibco, Grand Island, NY, USA) supplemented with 10% fetal bovine serum (FBS) (Gibco, Grand Island, NY, USA) and stored in a 5% CO_2_, 37 °C incubator.

### 2.2. Primers, Probes, and Plasmid

Three sets of primers and probes were designed in this study, two of which were against *gH* (envelope glycoprotein H) and one of which was against *ICP8* (infected cell protein 8) of HSV-1. All primers and probes were synthesized by Sangon, Shanghai, China, with their sequences and fluorophores listed in Table 1. The gH plasmid was constructed by Sangon, Shanghai, China and contained the full-length sequence of the *gH* gene (Table 1).

### 2.3. Genome Extraction

The viral DNA genome was extracted and purified via the magnetic bead method with the Host Cell Residual DNA Sample Pretreatment Kit (SHENTEK, Huzhou, China), following the manufacturer’s instructions.

### 2.4. Droplet Digital PCR

We used the QX200™ AutoDG™ Droplet Digital™ PCR system (Bio-Rad, Hercules, CA, USA) to complete the assay. Briefly, the ddPCR mixture consisted of 1.1 μL of DNA template, primers, probe, and 2 × ddPCR Supermix for Probes (No dUTP) (Bio-Rad, Hercules, CA, USA). In the final 22 μL quantification reaction system, the final concentrations of primers and probes were 900 nM and 250 nM, respectively. The sample reaction system was added to a new 96-well plate with 8 wells in a row. We turned on the automated droplet generator and put these consumable materials and reagents into order: DG32 automated droplet generator cartridges, pipet tips for the AutoDG™ system, 96-well plate sample plate containing the ddPCR reaction system, a clean 96-well plate for receiving microdroplets, and the automated droplet generation oil for probes. We ran the droplet generator system, sealed the 96-well plate containing microdroplets with pierceable foil heat seals, and ran the PCR system in any PCR machine. The PCR procedure was as follows: 95 °C for 10 min; 94 °C for 30 s, 60 °C for 60 s, and 98 °C for 10 min. This was repeated for 41 cycles, then held at 12 °C. Finally, the data was read in droplet reader with QuantaSoft (Bio-Rad, Hercules, CA, USA).

### 2.5. Production of oHSV Samples

Bioreactor was used for the oHSV production. Briefly, Vero cells were cultured with M199 (Gibco, Grand Island, NY, USA) supplemented with 5% fetal bovine serum in bioreactors. When the glucose achieved the optimal concentration, the medium was drained, and cells were rinsed with M199 medium once, prior to the addition of new M199 medium. oHSV was then added at a multiplicity of infection (MOI) of 0.03. Cytopathic effects were frequently monitored, and supernatants were harvested every 12 h, starting from 72 h post infection (hpi), until complete CPE was observed. All the harvested supernatants were clarified, concentrated, and purified. In each process, a small volume of the sample would be kept for virus titration and genomic copy number quantification.

### 2.6. Virus Titration

The plaque assay and CCID_50_ were used to quantify the infectious titer of the viruses. For plaque assay, 600 μL of 10-fold serially diluted viruses were added into the 6-well plate containing approximately 1 × 10^6^ Vero cells/well. After one hour of adsorption, the inoculum was removed and replaced with an overlay mixture of 1 mL of 1.2% agarose (Solarbio, Beijing, China) and 1 mL of 2 × DMEM medium supplemented with 5% FBS. The plate was placed back to the cell incubator for 3 days. The cells were fixed with 10% formalin (Beyotime, Shanghai, China) for 20 min prior to the removal of the overlay and staining with crystal violet (Beyotime, Shanghai, China). Plaques were counted, and the titer was calculated based on the plaque numbers in the dilution well and the initial inoculum used. For CCID_50_, Vero cells were pre-seeded in a 96-well plate at a density of 3 × 10^4^ cells/well. At the next day, the medium was removed, and 100 μL of 10-fold serially diluted viruses were added into each column of the 96-well plate, that is, one dilution for 8 replicates. After one hour of adsorption, an additional 100 μL of DMEM medium containing 5% FBS was added into each well. The plate was put back to the cell incubator for 5 days, followed by the counting of the wells, where cytopathic effects were observed. The virus titer was calculated using the Kärber method described elsewhere.

### 2.7. Statistical Analysis

The results and data of ddPCR were generated and analyzed by QuantaSoft software version 1.7.4.0917 (Bio-Rad, Hercules, CA, USA). Statistical analysis was performed using the GraphPad Prism software version 8.0.2 (GraphPad Software, San Diego, CA, USA). The significance was determined using *T*-tests.

## 3. Results

### 3.1. The Establishment of a Digital Droplet PCR (ddPCR) for Quantifying the Genomic Copy Number of oHSV-1 Samples

To accurately quantify the genome copies of oHSV-1, we first designed three sets of primer pairs and probes, two of which were against *gH* (envelope glycoprotein H) and one of which was against *ICP8* of oHSV-1. The sequences of primers and probes used in this study are listed in Table 1.

The oHSV-1 DNA genome was first extracted and serially diluted (16-, 32-, 64-, 128-, 256-, 512-, 1024-, 2048-, and 4096-fold) prior to the detection by the ddPCR with different sets of primers and probes. As shown in Figure 1A–C, all sets of primers and probes showed a good linearity between the genomic copy number and the inverse of the dilution ratio (R^2^ > 0.99). In addition, there was little difference regarding the genomic copy numbers when different primer pairs were used to detect samples at each designated dilution (*p* > 0.05) (Figure 1D). It indicated that the results of ddPCR with the three sets of primers and probes were accurate and stable. In terms of the fold of dilution prior to the ddPCR detection, the relative standard deviation (RSD) was relatively small when the samples were 16- to 256-fold diluted, and the RSD was the smallest when the samples were 128- and 256-fold diluted, suggesting that the dilution of the sample to the appropriate ratio contributed to achieving more accurate results (Table 2). Therefore, the gH-1 primer pair and probe were chosen for the experiments thereafter.

### 3.2. Specificity of the ddPCR Assay for Detecting oHSV-1

To verify the specificity of the established ddPCR assay, gH-1 primers and probes were used to amplify the genomes of oHSV-1, HSV-2, and the varicella zoster virus (VZV), which all belong to *Alphaherpesvirinae*. The genomes of oHSV-1, HSV-2, and VZV were extracted and detected by the ddPCR. The result showed that the genomic copy number of the purified oHSV-1 was 2.16 × 10^7^ copies/μL, while a positive signal was barely detectable for HSV-2 and VZV (Figure 2C). The amplification of the medium and PBS, which were used during the harvest and purification process of the oHSV, led to no positive signal as well. These results confirmed the high specificity of the ddPCR assay with the primers and probes of *gH*.

### 3.3. The Plasmid DNA and Limit of Detection of the ddPCR Assay

The gH plasmid DNA was serially diluted for the detection of the genomic copy number by ddPCR. As shown in Figure 3A, there was good linearity between the logarithm of the genomic copy number and the logarithm of the dilution ratio, with the linear correlation coefficient, R^2^, being 0.9936. This suggested that circular DNA could also be suitable for detection by ddPCR.

To identify the lower limit of quantification (LOQ) of the ddPCR assay, the genomic copy number of oHSV-1 DNA was detected (Figure 3B). Then, the DNA was diluted and added to the ddPCR mixture (20 μL) at a final concentration of 100 or 10 copies/μL. It showed that the positive signal was clearly distinguished for the 10 copies/μL sample, compared to the negative controls, indicating that the LOQ was 10 copies/μL (Figure 3E). The increase in volume in template DNA may effectively improve the minimum of the genomic copy number detection of the samples.

### 3.4. The Accuracy and Reproducibility of Genomic Copy Numbers of Samples Using ddPCR

To further confirm the reproducibility of the ddPCR assay, the same experiment was performed for a total of four times by two researchers on different days. A good linear correlation was found between the logarithm of the genomic copy number and the logarithm of the dilution ratio in each experiment, where there was little difference in terms of the slopes and intercepts of the linear equations (Figure 4). This indicated that the reproducibility of the established ddPCR assay for the detection of the genomic copy number was well-verified.

To verify the accuracy of the ddPCR detection, the genomic copy number of the serially diluted viruses was determined by quantitative PCR (qPCR) and ddPCR in parallel, and the infectious titer was measured by plaque assay as well. The results showed that there was a good linear correlation between the logarithm of the infectious titer and the logarithm of the dilution ratio (Figure 4E). Meanwhile, good linearity was also observed between the logarithm of the genomic copy number and the logarithm of the dilution ratio detected by qPCR and ddPCR, respectively (Figure 4F,G). Notably, the slopes of the linear equations obtained by the three methods were very similar to each other (Figure 4E–G). To evaluate the accuracy of ddPCR, the linear dependence of the logarithm of the genomic copy number by qPCR and ddPCR with the logarithm of the infectious titer was calculated, and good linear correlations were observed, with similar slopes of 0.9869 and 0.9578, respectively (Figure 4H,I). Then, the genomic copy number of the samples diluted 125-fold and 250-fold were compared between ddPCR and qPCR, and the results showed that there was no significant difference in the genomic copy number of the samples diluted at the same fold detected by the two methods (Figure 4J).

### 3.5. The Application of the ddPCR Assay during the oHSV Production

We next employed the ddPCR assay to detect the oHSV genomic copy number during virus production to monitor the virus titer for the best harvest time point. oHSV was added to the bioreactor at an MOI of 0.03 under three different conditions (batches S1~S3). Starting from 72 h post infection (hpi) of oHSV, supernatants were harvested for the genomic copy number detection by ddPCR every 12 h until complete cytopathic effects (CPE) were observed. As shown in Figure 5, the batch S1 had the highest genomic copy number at 120 hpi, while batches S2 and S3 yielded much lower genomic copy numbers than the S1 batch. Meanwhile, we quantified the virus titers for all harvested supernatants using the classical plaque formation unit assay. The results were well-associated with the results from the ddPCR assay, indicating that the latter could be a good alternative to monitor the virus titer to obtain the best yield during virus production.

### 3.6. The ddPCR Assay Could Accurately Detect the Genomic copy number of oHSV-1 without Viral DNA Extraction

The DNA extraction is usually necessary before PCR/qPCR. However, this potentially causes errors, especially when accurate quantification is required, due to a suboptimal recovery rate or technical mistakes during the DNA extraction. It has been extensively demonstrated that the ddPCR assay is barely affected by the various inhibitors, which allows the accurate quantification of samples without the DNA extraction. Therefore, we further investigated if the ddPCR could quantify the oHSV-1 genome directly without the DNA extraction. As presented in Figure 6C, there was hardly any noticeable difference regarding the genomic copy number between the extracted DNA and the purified virus without DNA extraction. Notably, although the distinction between positive and negative droplets was relatively indistinct (Figure 6A), the threshold line close to the negative droplets ensured that the genomic copy number obtained from the purified virus without DNA extraction was comparable to that of the extracted DNA.

### 3.7. The Genomic Copy Numbers of the Harvested and Purified Virus Showed Good Linear Relationships with Viral Infectious Titer

It usually takes 2–4 days for virus titration by CCID_50_ or plaque assay, so it would be very convenient to use the genomic copy number to calculate the viral infectious titer using a certain formula. For this purpose, we measured the virus infectious titer by plaque assay and the genomic copy number of the harvested and purified virus without extraction of viral DNA by ddPCR (Figure 7A–D). The results showed that there were good linear relationships between genomic copy numbers and the viral infectious titer. After the DNA extraction of the purified virus, the genomic copy numbers also showed a good linear relationship with the viral infectious titer (Figure 7E,F). In order to reflect the linear relationship more truly between the genomic copy number and the infectious titer, the viral infectious titers were detected by CCID_50_ or plaque assay after multiple dilutions, and the genomic copy numbers were detected at the same time by ddPCR. The results showed that the genomic copy numbers also showed good linear relationships with the viral infectious titer (Figure 7G,H).

## 4. Discussion

OVs are wild-type or genetically modified viruses that can selectively replicate in and kill cancer cells while leaving normal cells unharmed [33]. OVs represent a novel and promising immunotherapeutic method for the treatment of cancer, with the milestone of the FDA’s approval of Talimogene laherparepvec (T-VEC) from Amgen for the treatment of melanoma in October 2015 [34]. Due to the great enthusiasm from scientists and the rapid development of technologies, a wide diversity of viruses have been studied for their potentials as oncolytic viruses, including adenovirus, poxvirus, herpesvirus, rhabdovirus, reovirus, paramyxovirus, parvovirus, and picornavirus [35,36] In the last three decades, HSVs have become one of the best and the most extensively studied OVs, due to its broad cell tropism, potent oncolytic activity, large genome for the great compacity of inserted transgenes, and good safety profile. Indeed, of the four OV products that have been approved around the world, two are modified HSVs, namely the T-vec from Amgen and the Delytact from Daiichi Sankyo of Japan. Similarly, among the OV-based clinical trials worldwide, HSV-based products account for approximately 20%. Surprisingly, this number increased to 61% in China [37]. As a result, numerous oHSV products have been developed and investigated in phases I and II clinical trials by different pharmaceutical companies, such as RP1 of Replimune, ONCR-177 of Oncorus [38], MVTR-3011 of Immvira [39], and VG161 of Virogin Biotech [40].

Despite the promising prospective, one of the big challenges for pharmaceutical companies is the large-scale production of the clinical-grade virus of high titer. Generally, the manufacturing process of live oHSV products includes cell expansion, virus infection and production, harvest of the supernatant from infected producer cells, recovery, purification by size exclusion chromatography and/or ion-exchange chromatography, nuclease treatment to remove contaminating DNA, sterile filtration, and filling of the final product [41]. The viral titer is the most critical factor and usually measured in almost each processing stage. Therefore, accurate titration is of great importance for oHSV manufacturing. In this study, we established a ddPCR assay for the quantification of the HSV-1 titer and applied it into our oHSV manufacturing. The ddPCR assay exhibited great specificity, linearity, reproducibility, and accuracy (Figure 1, Figure 2, Figure 3 and Figure 4). We employed it to explore the optimal harvest time by measuring viral titers of different time points, which yielded similar but much faster results, compared to the plaque assay and CCID_50_. To further confirm the association between the ddPCR and the two methods, we compared the three methods by measuring serially diluted samples. A good linear correlation was found between the genomic copy number and the infectious titer by CCID_50_ and plaque assay, suggesting that ddPCR could be used as an alternative and fast method for quantifying the virus. This greatly accelerated the optimization of the conditions during the virus production. In addition, as a critical quality control category, the ddPCR assay was used to assess the titer of the purified virus, which is the active substance for the product, which demonstrated rapid and reliable results. Together, ddPCR substantially facilitated the oHSV’s manufacturing process.

The stability of the OV drugs is another important category that should be assessed regularly and precisely. The traditional plaque assay and CCID_50_ are subject to relatively low reproducibility due to the subjectivity and technical errors during each experiment. This may yield inaccurate or misleading results for the stability of the product, especially considering that the duration of a stability study can be up to one year or even longer. In the present study, two individuals performed the ddPCR for the quantification of the viral genomic copy number, and each individual performed it twice on two different days. A good linear correlation was found between the logarithm of the genomic copy number and the logarithm of the dilution ratio in all four experiments, with fairly close slopes and intercepts of the linear equations (Figure 4). Moreover, there was no significant difference in the genomic copy number detected by qPCR and ddPCR (Figure 4). Additionally, the ddPCR demonstrated great reproducibility through independent experiments by different researchers. Together, ddPCR would be a reliable tool to be employed for the stability study of our oHSV products at each processing stage. On the other hand, we evaluated that the LOQ of the assay was 10 copies/μL (Figure 3), which is useful for the early diagnosis of HSV-1 infection.

Moreover, it is worth noting that in this study, we compared the quantification results of the extracted viral DNA and purified oHSV virus without genome extraction. It was found that when the purified virus was directly measured by ddPCR, the positive and negative partitions were not well-separated. For a ddPCR assay, a threshold is required to distinguish the positive from the negative partitions, which directly affects assay validity and accuracy. In the present study, despite the poor demarcation, the threshold line automatically set by the instrument was close to the negative partition, ensuring the reliability of the assay. As a result, the genomic copy number measured from the purified virus was slightly higher than that of the extracted DNA. This might be due to the suboptimal recovery of the DNA from the extraction. Therefore, measuring the purified virus directly by ddPCR is not only more convenient and saves more time, it also is more likely to represent the actual result.

## 5. Conclusions

In summary, ddPCR is a great way to quantify HSV-1 and oHSV production, as it showed good specificity, linearity, low limit of quantification, great reproducibility, and accuracy. Moreover, ddPCR presented good resistance against PCR inhibitors. The ddPCR could be used to monitor the viral titer in real time during the production of oncolytic HSV (oHSV) for the determination of the optimal harvest time and to facilitate the manufacturing process of oHSV products.

## Figures and Tables

**Figure 1 viruses-15-00178-f001:**
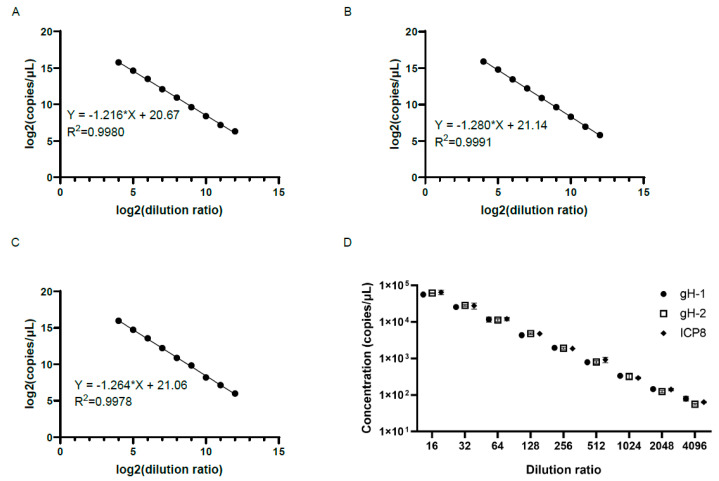
Detection with different sets of primers and probes by the ddPCR (**A**,**C**). There was a good linearity between the logarithm of the genomic copy number and the logarithm of the dilution ratio when using primer pairs gH-1 (**A**) and gH-2 (**B**) of the *gH* gene and ICP8 (**C**) primer pairs of the *ICP-8* gene for detection. (**D**) There was little difference in genomic copy numbers when using different primer pairs to detect samples with the same fold dilutions (*p* > 0.05). The *x*-axis represents the dilution ratio, and the *y*-axis represents the copy numbers of samples. The genomic copy numbers using different primer pairs to detect samples with the same fold dilutions were compared using T test.

**Figure 2 viruses-15-00178-f002:**
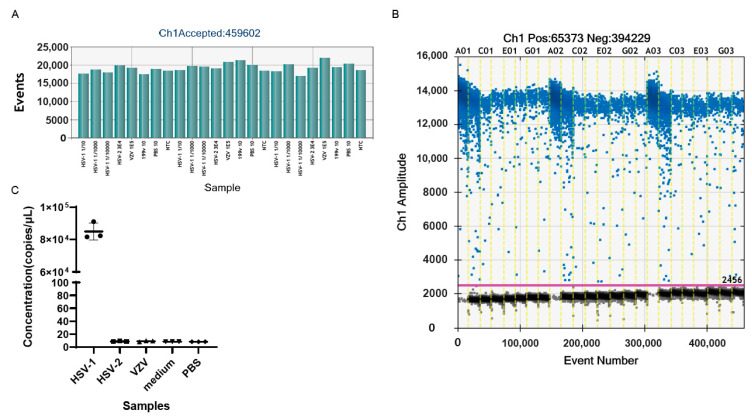
The specificity of gH primers and probes. (**A**) The total number of droplets of samples generated by ddPCR in the experiment of (**C**). The numbers exceeded 10,000, indicating that the experimental results were credible. (**B**) The one-dimensional (1D) amplitude of the samples in the experiment of (**C**). The positive droplets were labeled with blue dots, while the negative droplets were labeled with black dots. The threshold was indicated by the pink line. (**C**) The genomic copy numbers of oHSV-1, HSV-2, VZV, medium, and PBS. Every dot in the figure represents a technical replicate. The primers and probes used in the above experiments were gH-F1, gH-R1, and gH-P1. Evens: the total number of droplets. Ch1 amplitude: the fluorescence intensity of every droplet. NTC: no template control.

**Figure 3 viruses-15-00178-f003:**
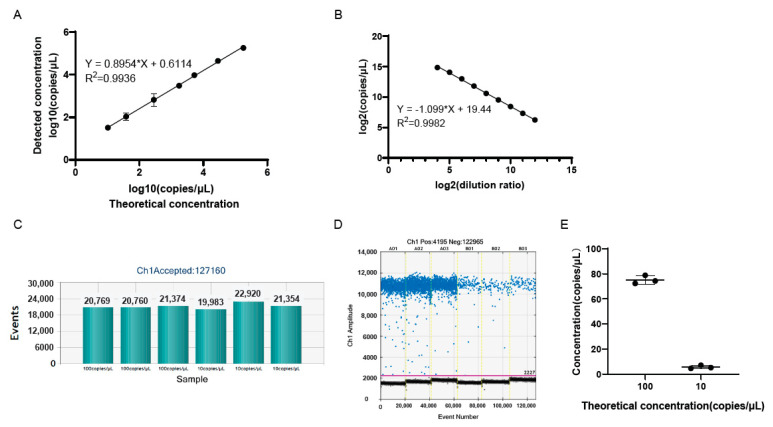
The linear correlation and minimum copy number detection. (**A**) The linear correlation between the logarithm of the copy number of gH plasmid DNA detected by ddPCR and calculated according to plasmid DNA concentration. The copy number of plasmid DNA = [6.02 × 10^23^ × plasmid DNA concentration(ng/μL) × 10^−9^]/(3025 × 660). (**B**) The linear correlation between the logarithm of the genomic DNA copy number of the samples with the logarithm of the dilution ratio. (**C**) The total droplet numbers of samples, indicating the validity of the results of the experiment in (**E**). (**D**) The 1D amplitude of the samples in the experiment of (**E**). The positive and negative droplets were labeled with blue and black dots, respectively. The threshold was indicated by the pink line. (**E**) The genomic copy number of low-copy samples were detected by ddPCR. Every dot in the figure represents a technical replicate. The *x*-axis represents the theoretical genomic copy number after the sample was diluted. The *y*-axis represents the genomic copy number detected by ddPCR.

**Figure 4 viruses-15-00178-f004:**
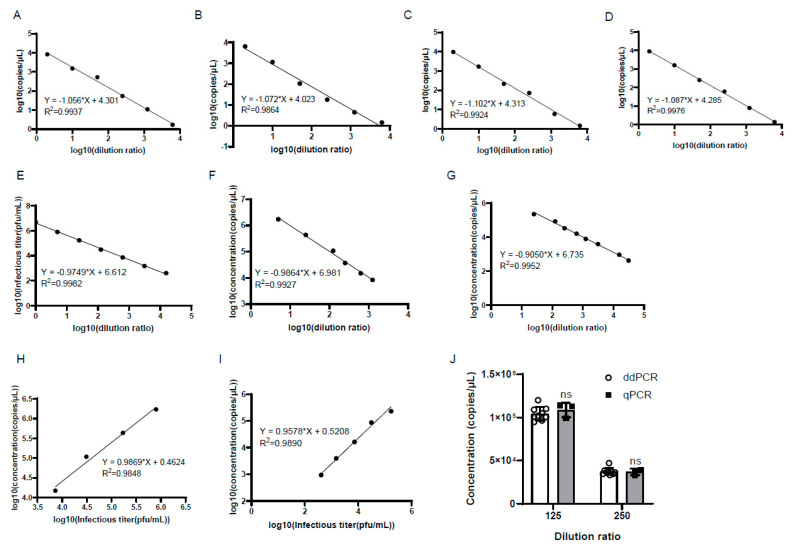
There was good repeatability in the genomic copy numbers of the samples by ddPCR detection. (**A**,**B**) The experiments were performed by one researcher on different days. (**C**,**D**) The experiments were repeated by another researcher on different days. The *x*-axis represents the logarithm of the dilution ratio, and the *y*-axis represents the logarithm of the genomic copy number detected by ddPCR. The DNA samples were 5-fold serially diluted. (**E**) There was a good linearity between the logarithm of the infectious titer and the logarithm of the dilution ratio. The infectious titer was detected by plaque assay. (**F**,**G**) There were good linearities between the logarithm of the genomic copy number by qPCR and ddPCR and the logarithm of the dilution ratio, respectively. (**H**,**I**) Good linear correlations were found between the logarithm of the genomic copy number by qPCR and ddPCR and the logarithm of the infectious titer, respectively. (**J**) There was no significant difference in the genomic copy number by qPCR and ddPCR when the samples were diluted at the same fold; ns: not significant.

**Figure 5 viruses-15-00178-f005:**
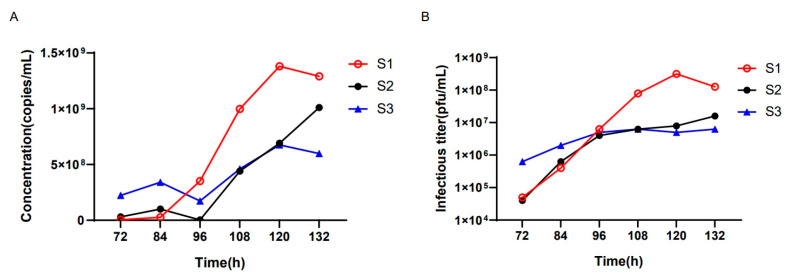
The genomic copy number of oHSV-1 in different hours cultured in a bioreactor. (**A**) The culture conditions were different in the S1, S2, and S3 batches. The *x*-axis represents different hours after oHSV-1 infection, and the *y*-axis represents the DNA copy numbers per milliliter. (**B**) The viral infectious titers under different culture conditions were measured. The *x*-axis represents different hours after oHSV-1 infection, and the *y*-axis represents the infectious titer.

**Figure 6 viruses-15-00178-f006:**
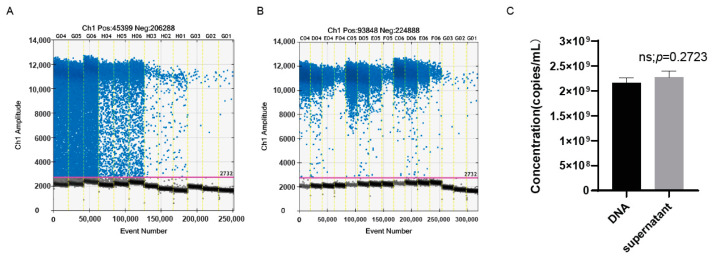
The genomic copy number measured without the extraction of viral DNA was comparable to that of the extracted DNA. (**A**) The 1D amplitude of the samples without the extraction of viral DNA. The positive and negative droplets were labeled with blue and black dots, respectively. The threshold was indicated by the pink line. (**B**) The 1D amplitude of the samples after the extraction of viral DNA. (**C**) The genomic copy numbers per milliliter of the purified virus with or without the extraction of viral DNA. DNA: detection after the extraction of viral DNA; supernatant: detection without the extraction of viral DNA.

**Figure 7 viruses-15-00178-f007:**
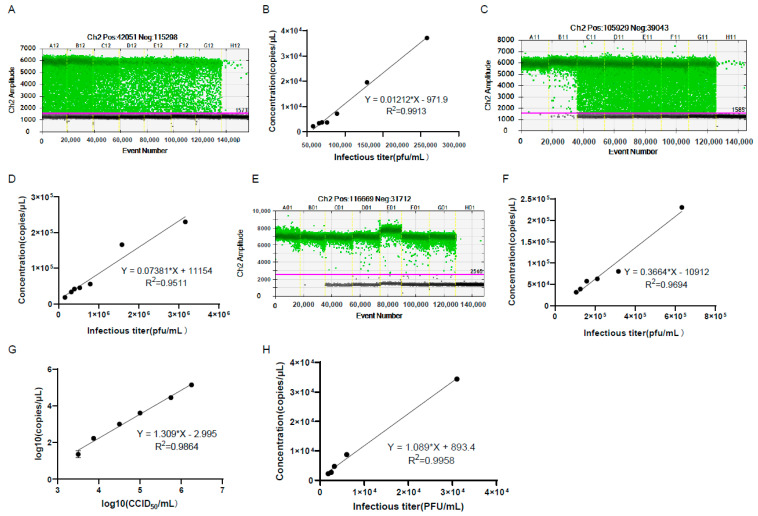
The genomic copy number of the harvested and purified virus showed good linear relationships with the viral infectious titer. (**A**–**C**) shows the 1D amplitude of the harvested (**A**) and purified virus (**C**) in genomic copy number detection in (**B**–**D**), respectively. The positive and negative droplets were labeled with green and black dots, respectively. The threshold was indicated by the pink line. The genomic copy number of the harvested (**B**) and purified virus (**D**) showed good linear relationships with the viral infectious titer. The viral infectious titers were detected before virus dilution. (**E**) The 1D amplitude of the purified virus after the extraction of virus DNA in the experiment of (**F**). (**F**) The genomic copy number of the purified virus DNA showed good linear relationships with the viral infectious titer. The genomic copy number of the purified virus DNA showed good linear relationships with the viral infectious titers detected by CCID_50_ (**G**) or plaque assay (**H**). The viral infectious titers were detected after multiple dilutions of the virus.

**Table 1 viruses-15-00178-t001:** The information of primers and probes.

Name	Sequence
gH-F1	GGCTGCGTGTCAAAGGCTA
gH-R1	GTGTTTTCGAGCGACGTGC
gH-P1	5′6-FAM-ACGGCCTTGTTGCTATTTCCAAA-3′BHQ1
gH-F2	CGGCGATTTGCTGCTGT
gH-R2	ACCCGGGATATCGAGTCCAA
gH-P2	5′HEX-CACCCAAAACCAGCGCGACCT-3′BHQ1
gH plasmid DNA	CGGGCTGCGTGTCAAAGGCTAGCAAATGAATGACGGTTCCGTTTGGAAATAGCAACAAGGCCGTGGACGGCACGTCGCTCGAAAACACGCTCGGGGCGCCCTCCGTCGGCCCGGCGGCGATTTGCTGCTGTGTGTTGTCCGTATCCACCAGCAACACAGACATGACCTCCCCGGCTGGGGTGTAGCGCATAAACACGGCCCCCACGAGCCCCAGGTCGCGCTGGTTTTGGGTGCGCACCAGCCGCTTGGACTCGATATCCCGGGTGGAGCCTTCGCATGTCGCGGTGAGGTAGGTTAGGAACAGTGGGCGTCGGA
ICP8-F	TCGCCACGAACACGCTACT
ICP8-R	CCCGCTCCTTATTTTTGACC
ICP8-P	5′6-FAM-CCCGCCCTCGGAGATAATGC-3′BHQ1

**Table 2 viruses-15-00178-t002:** Results of genomic copy numbers with different primer pairs under same fold dilutions.

Primer Pair	Dilution Ratio								
	16	32	64	128	256	512	1024	2048	4096
gH-1	55,400	25,980	10,440	4220	1932	798	348	152	88
	57,220	25,340	13,080	4500	1982	788	324	138	72
gH-2	60,560	27,560	10,640	4660	1834	850	360	130	54
	62,860	29,660	11,900	4940	1994	756	286	118	58
ICP8	71,000	31,540	13,100	4760	1868	1040	304	134	66
	58,560	24,040	11,240	4760	1880	814	284	150	62
Average	60,933.33	27,353.33	11,733.33	4640.00	1915.00	841.00	317.67	137.00	66.67
RSD	8.35	9.42	9.09	4.94	3.09	11.10	9.16	8.50	16.67

## Data Availability

Not applicable.

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
