# Peer review of "Development and Preliminary Application of a Droplet Digital PCR Assay for Quantifying the Oncolytic Herpes Simplex Virus Type 1 in the Clinical-Grade Production"

_viruses, 2023, doi:10.3390/v15010178_

Round 1

Reviewer 1 Report

I think the authors are doing a great job for developing a droplet digital PCR assay for quantifying oncolytic Herpes Simplex Virus type 1.

For establishing lower limit of quantitation (LLOQ), the authors might want to do that based on the noise (at least five times higher), which is more convicing.

Minor issues:

in line 55, please revise “viral acid nucleic”.

In line 190, I don’t see figure 2E for fig 2.

Reviewer 2 Report

1.This research focused on Development and preliminary application of a droplet digital PCR assay for quantifying oncolytic Herpes Simplex Virus type  1 in clinical-grade production, after check the pubmed,this work was so very prospective.

2.DDPCR looks more effectvie, but Figure 4 revealed that no diffence between DDPCR and qPCR.

3. Logic analysis in Figure 1 3 7 I think should showed the R value and P value?

4. Figures not very clear need much more than 300 dpi.

Round 2

Reviewer 2 Report

thanks very much your reply,firstly I  want to show my respect that you have done good work,this study let readers including me to understand the latest scientific research trends; Secondly I want to show my apologize that I have misunderstanded  the Figure 1 3 7 Statistical methods, you are right, this is  linear analysis like standard curve is required for protein concentration determination, not necessary or can not to Statistical P value; I have made mistake this was correlation analysis. At last I endorsed publication of this manuscript.